# Decompressive Hemicraniectomy without Evacuation of Acute Intraparenchymal Hemorrhage

**DOI:** 10.3390/biomedicines12081666

**Published:** 2024-07-25

**Authors:** Cristóbal Blanco-Acevedo, Eduardo Aguera-Morales, Antonio C. Fuentes-Fayos, Nazareth Pelaez-Viña, Rosa Diaz-Pernalete, Nazaret Infante-Santos, Ana Muñoz-Jurado, Manuel F. Porras-Pantojo, Alejandro Ibáñez-Costa, Raúl M. Luque, Juan Solivera-Vela

**Affiliations:** 1Department of Neurosurgery and Neurology, Reina Sofia University Hospital (HURS), 14004 Cordoba, Spain; doctoredu@gmail.com (E.A.-M.); nazaret1989vega@gmail.com (N.P.-V.); juan.solivera@gmail.com (J.S.-V.); 2Maimonides Institute of Biomedical Research of Cordoba (IMIBIC), Reina Sofia University, Hospital University of Cordoba, 14004 Cordoba, Spain; b22fufaa@uco.es (A.C.F.-F.); b22mujua@uco.es (A.M.-J.); aibanezcosta@gmail.com (A.I.-C.); bc2luhur@uco.es (R.M.L.); 3Department of Medical and Surgical Sciences, University of Cordoba, 14004 Cordoba, Spain; 4Department of Cell Biology, Physiology and Immunology, University of Cordoba, 14014 Cordoba, Spain; 5CIBER Physiopathology of Obesity and Nutrition (CIBERobn), 14004 Cordoba, Spain; 6Intensive Care Service, Reina Sofia University Hospital (HURS), 14004 Cordoba, Spain; rosa_mcdp@hotmail.com (R.D.-P.);; 7Neurosurgery Service of the Hospital del Mar, 08003 Barcelona, Spain; nazaret.infante@gmail.com

**Keywords:** intracerebral hemorrhage, intraparenchymal hematoma, decompressive craniectomy, clot evacuation

## Abstract

Background: Intracerebral hemorrhages (ICHs) are prevalent, with high morbidity and mortality. We analyzed whether decompressive craniectomy (DC) without evacuation of the acute intraparenchymal hematoma could produce better functional outcomes than treatment with evacuation. Methods: Patients with acute ICH treated with DC without clot evacuation, or evacuation with or without associated craniectomy were included. Matched univariate analyses were performed, and a binary logistic regression model was constructed using the Glasgow Outcome Scale (GOS) and modified Rankin scale (mRS) as dependent variables. Results: 27 patients treated with DC without clot evacuation were compared to 36 patients with clot evacuation; eleven of the first group were matched with 18 patients with evacuation. A significantly better functional prognosis in the group treated with DC without clot evacuation was found. Patients aged < 55 years and treated with DC without clot evacuation had a significantly better functional prognosis (*p* = 0.008 and *p* = 0.039, respectively). In multivariate analysis, the intervention performed was the greatest predictor of functional status at the end of follow-up. Conclusions: DC without clot evacuation improves the functional prognosis of patients with acute intraparenchymal hematomas. Larger multicenter studies are warranted to determine whether a change in the management of acute ICH should be recommended.

## 1. Introduction

Intracerebral hemorrhage (ICH), a subtype of stroke, is a devastating and common condition, in which a hematoma forms within the brain parenchyma with or without blood spread to the ventricles [1]. ICH accounts for 10 to 15% of all strokes [2] and is associated with high morbidity and mortality [3]. It has a hospital mortality rate of 40% [4,5] and one year after the hemorrhage, 75% of patients are severely disabled or die [5,6].

Hemorrhages within the brain parenchyma are often classified into primary injury, i.e., tissue injury caused immediately by the hematoma, and secondary injury, which involves a subsequent pathological change resulting from the hemorrhage [1]. Intracerebral hemorrhages (ICHs) are prevalent, and morbidity and mortality are high. Medium (>30 mL) and large hematomas (>60 mL) represent a particular challenge, due to the increase in intracranial pressure (ICP) and its association with a worse prognosis [7,8]. In this regard, it has been determined that hematoma volume is a good predictor of mortality at 30 days [9]. In addition, there are neurological complications of ICH, including hematoma expansion, perihematomal edema, intraventricular extension of hemorrhage, hydrocephalus, seizures, and non-neurological complications, such as aspiration pneumonia and venous thromboembolism, that also lead to a bad prognosis [10].

Despite improvements in ICH treatments and prevention, there has been little improvement in mortality over the past 30 years [4]. In general, the treatment of patients with ICH remains controversial [11] and an unsolved problem, since, apart from a care bundle protocol, all pharmacological and surgical treatment approaches have failed to reduce morbidity and mortality [12].

Decompressive craniectomy (DC) is a procedure that reduces intracranial pressure and improves prognosis in patients with malignant stroke. However, its usefulness in ICH is unclear [13]. Various studies have shown that decompressive craniectomy without clot evacuation has lower mortality and is associated with a better functional outcome [12,13], although it may present some risks such as hemorrhagic complications [12].

For all this and due to the discouraging outcomes in the quality of life and functional response of patients undergoing medical and/or surgical treatment with clot evacuation [14], the objective of this comparative study was to analyze the viability of DC without clot evacuation in terms of survival and functionality.

The foundation of this study was to obtain two homogenized groups based on their clinical and radiological characteristics and manner of action (time until surgical treatment), which allowed us to have an approximation of the management and functional prognosis in moderate and large-volume brain hemorrhages.

## 2. Materials and Methods

### 2.1. Human Cohort

We conducted an observational retrospective cohort analysis comparing two historical series of 63 patients with acute IPH and candidates for surgical treatment. A total of 27 patients underwent DC without clot evacuation. On the other hand, for 36 patients, the surgical approach was craniotomy and/or DC with clot evacuation. Patients were then matched by clinical and radiological characteristics (*n* = 29) and divided into 2 groups. Thus, according to clinical and radiological characteristics, we selected and matched 11 patients with acute IPH who underwent DC without clot evacuation between 2010 and January 2020, and 18 patients from the retrospective series who had undergone clot evacuation since 2007. The median follow-up was 7 months. Patient follow-up was 1 to 120 months (mean ± 30 months standard deviation [SD]). In addition to the above, the patients were homogenized based on their age, location of the hematoma, and surgery time.

### 2.2. Patient Selection

The medical records of all patients treated in our unit since 2007 were reviewed, and all cases of patients treated with DC with or without clot evacuation were selected, excluding those with extra-axial hematomas (associated epidural and/or subdural hematomas).

Inclusion criteria were previous good functional status (Karnofsky Performance Status score > 80), presence of clinical or radiological signs of intracranial hypertension, signs of compression and/or midline shift at the level of the foramen of Monro [15,16], association with a hematoma volume > 30 mL measured by the ellipsoid volume (A × B × C)/2 [17], no evidence of pupillary alteration prior to treatment, time of surgical indication since the stroke (minor < 24 h), and signs of elevated intracranial pressure, such as headache, vomiting, and/or altered consciousness.

Exclusion criteria were advanced signs of herniation since admission, comorbidity predicting limited life expectancy, Karnofsky Performance Status score < 70 prior to surgical treatment, presence of comorbidities (such as cancer or effects on other organs or systems including trauma or other diseases that could modify the patient’s condition), as well as severe pre-existing neurological deficits, as the potential for significant recovery may be limited.

Patients with scores of 13 to 15 points on the Glasgow Coma scale (GCS) were not initially considered candidates for neurosurgical treatment.

### 2.3. Considerations for Decompressive Craniectomy without Clot Evacuation

The diameter of the craniectomy was at least 10 cm, and sufficiently large to allow a distance of at least 3 cm between the herniated parenchyma and the bone margins. The arachnoid membrane was not punctured to prevent cerebrospinal fluid leakage, in order to minimize changes in its dynamics. An intraparenchymal ICP sensor (Camino^®^, Integra^®^, Plainsboro, NJ, USA) was placed on the affected side when possible or, if the craniectomy was very close to the midline, on the contralateral side. At the end of the procedure, pressures below 6 mmHg were taken as an indication of adequate decompression.

### 2.4. Clot Evacuation

Lesion-centered craniotomy was performed, and the hemorrhage was evacuated using suction, preserving the surrounding brain as much as possible and avoiding the use of a brain spatula or extensive bipolar coagulation. In some cases, DC was also performed to help reduce ICP and improve postoperative management. In most patients, an intraparenchymal ICP sensor was also placed for monitoring in the intensive care unit.

### 2.5. Variables

The functional status of operated patients was assessed at the end of their follow-up using the Glasgow Outcome Scale (GOS) and modified Rankin Scale (mRS) [18,19]. The 1-month and 1-year mortalities were recorded. Independent variables recorded were age, sex, hematoma characteristics (etiology, laterality, location, volume, midline shift), GCS on admission, time from clinical deterioration to intervention, and type of treatment.

### 2.6. Statistical Analysis

In order to verify that the baseline characteristics of both groups were similar, all study variables were compared between groups. The Chi-square test was used for qualitative variables, with Yates correction if necessary. The Kolmogorov–Smirnov test was applied to check the normality of the continuous variables. In the case of normal variables, the Student’s *t*-test was used; otherwise, the Mann–Whitney U test was applied.

A binary logistic regression model was calculated to establish factors associated with the clinical outcomes. Functional recovery grouped according to the GOS or mRS was taken as a dependent variable (GOS 1–3 vs. GOS 4–5 or mRS 3–6 vs. mRS 0–2). The following were independent variables: treatment received, patient age as a categorical variable (≥55 vs. <55 years), sex, etiology, laterality, superficial or deep location, volume, midline shift, and GCS on admission. Using the Wald statistic, variables with *p* ≥ 0.15 (backward stepwise selection procedure) were eliminated from the model one by one. The reduced model was compared with the model that included the deleted variables using the likelihood-ratio test. Possible interactions between the variables were studied. Variables with statistical significances greater than 0.05 were studied as possible confounding factors and were taken as such if the percentage change of the coefficients was greater than 15%. Cook’s distance was used as a diagnostic test for extreme cases. The Hosmer–Lemeshow statistic, based on percentiles, was used to assess goodness of fit.

### 2.7. Ethical Concerns

All recruited subjects signed an informed consent form. This research followed the fundamental principles established in the Declaration of Helsinki, in the Council of Europe Convention on Human Rights and Biomedicine, in the Universal Declaration of UNESCO on human rights, and the requirements established in Spanish legislation in the field of medical research, such as Law 14/2007, of 3 July, on Biomedical Research and Organic Law 3/2018, of 5 December, on the protection of personal data, and bioethics and guarantee of digital rights. It also complies with the provisions of Law 31/1995, of 8 November, on Occupational Risk Prevention and the Royal Decrees that develop it, regarding risks related to exposure to biological agents. This study was approved by the Research Ethics Committee of Córdoba.

## 3. Results

### 3.1. Descriptive Statistics and Comparison of the Study Groups

A total of 18 patients with clot evacuation and 11 with DC without clot evacuation were included.

Table 1 shows the descriptive data and results of the univariate analysis between treatment groups. Approximately 80% of hemorrhages were spontaneous. Cases caused by severe head trauma, accounting for 20% of patients, were not excluded. The distribution of spontaneous and traumatic etiology was similar among the study groups. Both treatment groups were highly homogeneous in terms of demographic, clinical, and hematoma characteristics (Table 1 and Figure 1). No patient in the group treated with CD without evacuation underwent clot evacuation, while in up to 46% of patients in the evacuation group, some type of craniectomy with decompressive intent was performed. For this reason, the variables clot evacuation and DC were tested independently in the logistic regression model, although they were not significant.

Although nearly 80% of patients treated by DC without clot evacuation had hematomas with left laterality, compared to 46% in the evacuation group, the differences were not statistically significant.

The functional prognosis of patients evaluated by GOS or mRS, both pooled and by category, was statistically significantly better in patients who had undergone craniectomy without clot evacuation compared to the other group (*p* < 0.05; Figure 2).

None of the patients in the DC without clot evacuation group died during follow-up, while 23% of those who underwent clot evacuation died during the first 2 months of follow-up. In addition, 78% of patients with DC and without clot evacuation presented minimal disability or absence of significant disability in mRS, while only 15% presented minimal disability in the clot evacuation group and 54% had moderate to severe disability. Most patients in the CE without clot evacuation had moderate limitations (56%) or good recovery (22%) in the GOS, with significant differences (*p* < 0.05) with the CE group, in which 62% had severe limitations.

One-month survival of patients who underwent evacuation was 70%. This rate remained practically stable, falling to 67.5% at 6 months and to 64.9% at 1 year, while survival of the DC patients without evacuation was 100%. Although we observed a trend, this difference is not significant (*p* = 0.1; Figure 3).

### 3.2. Binary Logistic Regression

Table 2 summarizes the analysis of factors that confer a poor functional prognosis in patients treated for spontaneous hemorrhage in our series. In the univariate analysis, patients with clot evacuation had an up to 19-fold higher risk of having a poor functional prognosis (GOS 1–3 or mRS 3–6) compared to DC alone without evacuation (*p* < 0.05). The only two significant variables in the model were age as a categorical variable and treatment received. We decided to maintain age, although this parameter was insignificant in the final model (*p* = 0.100). The analysis did not reveal factors that might introduce bias in the prediction. No confounding factors or interactions were identified. Treatment received was the most important independent predictor of prognosis in patients treated for IPH. DC with or without evacuation was also independently evaluated as an independent variable, although it was not significant after including clot evacuation in the model (*p* = 0.906).

## 4. Discussion

In this study, subgroup analyses based on age or hematoma volume were not performed since the series of patients included in this study was limited to performing analyses in which conclusive results were obtained. Our main objective is to obtain homogenized groups based on their clinical and radiological characteristics to determine the functional prognosis and survival according to the most appropriate surgical technique.

The age of the groups was established as under 70 years of age, although initially the recommendations of the literature were followed that certain benefits of neurosurgical treatment can be found up to the age of 80. The impossibility of finding similar patients, at ages closer to 80 years, as well as younger patients, only allowed us to take into account the age of the study patients.

Regarding the volume of the hematoma, our study is based mainly on volumes of 30–60 mL. No patient with hematomas smaller than 30 mL or larger than 60 mL was included. In relation to those smaller than 30 mL, we think that medical treatment may be one of the best initial and most plausible alternatives. Hematomas of larger volumes have an unfavorable functional prognosis. In our opinion, the groups that benefit the most from surgical treatment are patients with hematoma volumes that are between 30 and 60 mL. For this reason, we try to clarify which of both surgical techniques can offer the best clinical and functional results.

### 4.1. Pathophysiological Considerations

The manner and rate of onset of ICH causes patients to become critical within a few hours [19]. No effective medical treatments are available yet to counteract the deleterious effects of intracranial hypertension on brain functions when hematomas reach considerable volumes, and the progression of mass effect is one of the mechanisms by which ICH can cause permanent brain damage [20]. The progression of mass effect has been well characterized; studies found that midline shift developed over the first 72 h, peaking within 3 to 5 days and remitting in approximately the next 14 days [21]. Early hematoma expansion and growth are more common, with up to 37% hematoma growth within the first 3 h of onset and up to 13.3% between 3 and 24 h [19,22,23]. However, it has also been observed that perihematomal edema increases by approximately 75% during the first 24 h after ICH [24], and does not decrease until 17 days later in small hemorrhages and even later—up to 28 days—in larger ones [25,26]. In this sense, it is worth noting that patients at our center are operated on within the first 24 h after the stroke.

The effect of open surgical evacuation of ICH has been widely discussed in the literature, including the randomized multicenter STICH study [27]. This trial found no difference in overall outcomes between surgical versus non-surgical treatment; however, a subgroup analysis indicated that patients with peripheral lobar hemorrhages could benefit from early surgical evacuation [14]. DC can be performed safely as part of the initial management of a subcategory of patients with subarachnoid hemorrhage (SAH) who present with large sylvian fissure hematomas [28]. In patients with severe brain injury, early DC without clot evacuation may be an option for reducing associated ischemic events, especially when ICP increases [29]. Recent experimental models have obtained significant reductions in mortality when craniectomy was performed early, within the first 24 h [30]. Likewise, the recent study carried out by [12] reported that DC did not increase the incidence or proportion of adverse events or serious adverse events and the duration of hospital stay was shorter in patients who received it. The combination of decompressive craniectomy with (partial) clot removal needs to be studied further in randomized controlled trials.

Therefore, our study aimed to assess and clarify whether there is a difference between these two surgical techniques, taking into account that the initial management of intraparenchymal hemorrhages with volumes > 30 mL should be a surgical indication as the first option. For those hemorrhages in which the hematoma volume is less than 30 mL, optimized medical treatment remains, in our opinion, the most plausible option. The volume of the hematoma is one of the most determining variables when indicating the treatment and initial management of intraparenchymal hemorrhage. Likewise, it is essential to take into account the comorbidities of each patient. In this sense, ICH is a pathology with great heterogeneity, so DC could be preferable in certain cases, although not in all. Therefore, treatment decisions must be individualized, taking into account, in addition to comorbidities, the size and location of the hemorrhage and the patient’s age and neurological status to determine whether to perform DC without clot evacuation.

### 4.2. Reason for Not Evacuating the Clot

The intrinsic mechanisms of the brain and its adaptive mechanisms after injury (cerebral hemorrhage) are increasingly being considered, even though it is difficult to demonstrate their leading role. This casts doubt on whether our way of acting is the most beneficial for the patients from a pathophysiological and functional prognosis point of view. For this, we list below the reasons that we consider most relevant:The discovery of neurogenesis in the adult brain suggested that in the central nervous system, as in other tissues, regeneration and cell replacement processes also occurred in pathological situations. For instance, after an acute insult such as a stroke, cell proliferation is stimulated [31,32].The vasculature plays an important role in long-term striatal neurogenesis after stroke. For several months, neuroblasts migrate close to the blood vessels through an area exhibiting early vascular remodeling. Optimization and preservation of vascularization should therefore be an important strategy for stimulating neurogenesis after stroke [33]. Studies showing vascular growth factors in the periphery of brain contusions [34] have generated debate as to whether these factors are responsible for trying to repair the damage caused by the stroke, whilst the risk of surgical manipulation in venous and arterial compromise in the brain has not been assessed.Another equally important concept is cerebral autoregulation, by which the cerebral vascular resistance is modified to maintain a cerebral blood flow (CBF) that meets the brain’s metabolic demand for oxygen at any particular time [35]. Concepts such as classic autoregulation (CA) [36,37] and dynamic autoregulation (DA) [38] have fostered progress in an area that is contributing enormously to the understanding of brain pathophysiology. CA and DA impairment is described in a number of pathological conditions, such as post-traumatic brain injury syndrome, SAH, acute ischemic stroke, and carotid vascular disease. It has been suggested that both CA and DA have different control mechanisms, and that DA is more susceptible to impairment in pathological situations but is unaffected by aging and the mechanisms underlying autoregulation, such as adrenergic mechanisms [39,40,41,42,43], cholinergic mechanisms [44], and myogenic modulation [45,46].

The use of DC early in the patient’s clinical course will reduce ICP [18], so we hypothesize that we might be acting before these adaptive mechanisms or adjustments have a chance to occur. Expanding the skull may delay and/or prevent pathophysiological mechanisms, such as cerebral edema, that can not only be attributed to the hematoma but also a possible sign of loss of cerebral compliance, producing unwanted effects on the correct functioning of the central nervous system.

One of the major limitations to conducting conclusive studies is the difficulty in creating similar study groups, given the wide heterogeneity of hemorrhages and underlying comorbidities of patients, including the number of hematomas, hemorrhage location, previous quality of life, use of antiplatelet or anticoagulant agents, presence of hematologic disease, or time from diagnosis to intervention [26]. Prior clinical assessment using scales may also bias patient selection, since the National Institutes of Health Stroke Scale (NIHSS) puts more emphasis on the left hemisphere than on the right. Additionally, the GCS may be understood very differently depending on whether the score for the motor response is low or high, taking into account that in most of these cases, patients display low levels of consciousness.

Publications on DC as the sole treatment for ICH have heightened interest in comparing clot evacuation with DC. The main argument for performing DC without clot evacuation is based on the immediate reduction in intracranial hypertension [47] and avoidance of unquantified damage from surgical manipulation and associated vascular compromise [13,45,48,49].

When an ICH is associated with moderate/severe traumatic brain injury, the mechanisms causing the lesion are likely to have a greater impact on the functional prognosis of the patient than in spontaneous ICH [29]. Furthermore, the presence of SAH due to a ruptured aneurysm may present a different functional prognosis than in ICH associated with amyloid angiopathy. However, we found that neither spontaneous nor traumatic etiology had an impact on the functional prognosis in the univariate analyses performed in our study, although a larger patient group would be needed to adequately assess the impact of these factors on functional prognosis.

Our data also support DC without clot evacuation rather than evacuation with or without craniectomy in patients with ICH with criteria for surgical treatment. The comparison showed that our treatment groups were well matched since there were no significant differences in their characteristics. The results of the univariate and multivariate analyses strongly suggest that clot evacuation results in a worse functional prognosis. The probability of mortality at 30 days after hemorrhage in patients with GCS > 9 and ICH volume of 30–60 mL was ~46%, while in those with GCS < 9 and the same volume range, the probability rose to 75%. Other authors note a high NIHSS score on admission, subarachnoid extension of the hemorrhage, and/or intraventricular hemorrhage as independent predictors [50]. In our study, we found no significant relationship between the hemorrhage size or GOS score and functional prognosis. The vast majority of the hemorrhages were medium or large, with a mean volume of 44 ± 16 mL, suggesting strict criteria for surgical indication. A systematic literature search concluded DC without clot evaluation may offer functional and heterogeneous studies [51]. However, a recent prospective randomized study found no statistically significant differences in the evacuation of deep hematoma or treatment by decompressive craniectomy without hematoma evacuation. However, it revealed a trend in functional prognosis in favor of hematoma evacuation [52]. Early DC with or without clot evacuation is feasible and safe for managing spontaneous ICH [11]. In another retrospective study, the performance of hematoma evacuation during decompressive hemicraniectomy for spontaneous intracerebral hemorrhage may not change functional outcomes over performing the hemicraniectomy alone [53]. The analysis of the results of these works reflects that there is no real consensus on this pathology.

### 4.3. Weaknesses and Strengths of the Study

This retrospective research has some limitations. This study has a small sample mainly due to the difficulty of finding “different types of hemorrhages” in each group and comparing them in age, location, and etiology, without the presence of an extra-axial component, which could alter the results. Furthermore, our study could suffer from selection bias, since patients with pupillary alterations were not included, due to the difficulty of establishing the time of onset until the time of surgery. Pupillary alterations are a relevant clinical sign that could significantly worsen the functional and survival prognosis. It is a variable to consider for future updates, as long as we can accurately verify the time from the appearance of this clinical sign of poor prognosis until the surgery is performed.

With regard to the limitations of the corresponding cases, a more prospective, randomized, multicenter study would be ideal so that the reproducibility of the results can be compared. The clinical evidence is currently unable to conclude on the most appropriate surgical technique. The scientific literature is mostly based on limited case series. However, the strength of this study lies in the fact that, over the last 15 years, the surgical technique has not been modified at our center and that both techniques are still in use today. Its indication and execution time were taken into account when homogenizing the groups.

There is no consensus in our center on when to apply one or the other technique. Likewise, there is literature that shows that surgical treatment prolongs the life and functional status of patients. However, this remains a matter of debate.

The prolonged evolutionary follow-up of our patients allows us to carry out follow-up clinical assessments for many years, which may be of interest to measure the evolutionary follow-up scales. The results obtained can contribute to the creation of an action protocol, considering that patients with intraparenchymal hematomas larger than 30 mL may require early neurosurgical treatment, without delaying the indication for other therapies.

## 5. Conclusions

DC without clot evacuation improves the functional prognosis of patients with acute IPH. The findings of our study support a change in the management of acute cerebral hemorrhage. Surgical treatment may be considered initially, as occurred with malignant middle cerebral artery infarction; however, the conduct of larger multicenter studies with greater statistical power will help to establish the indications for this technique more accurately. We believe that craniectomy should be indicated in medium- to large-volume hematomas, with the opening of the dura mater without clot evaluation, early in the patient’s clinical course and not be delayed by other therapies.

## Figures and Tables

**Figure 1 biomedicines-12-01666-f001:**
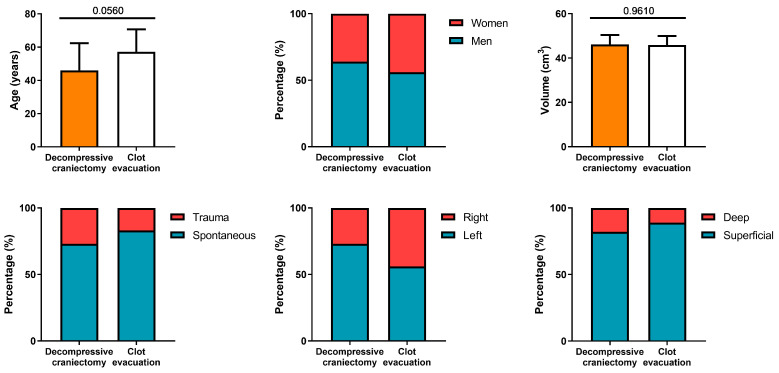
Comparison of demographic, clinical, and radiological characteristics of matched groups. No significant differences were found.

**Figure 2 biomedicines-12-01666-f002:**
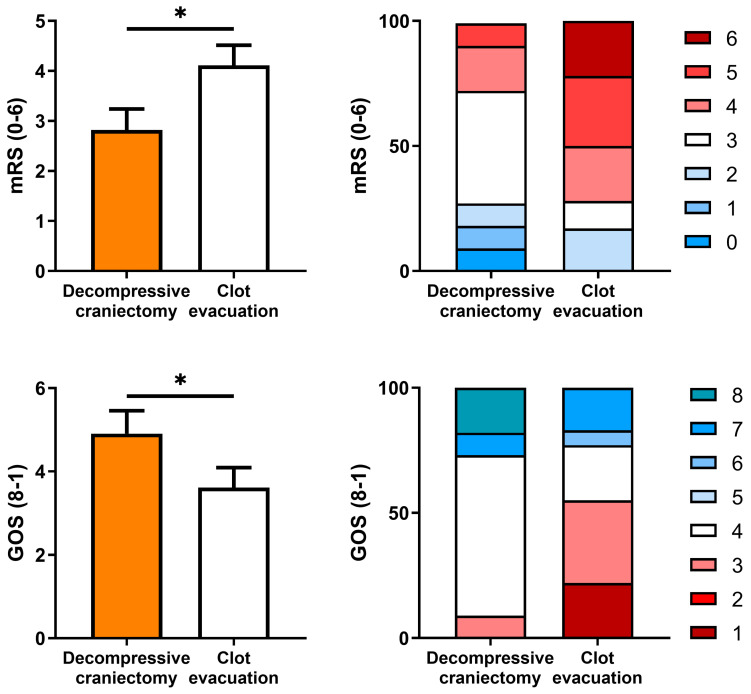
Clinical outcomes after follow-up of at least 6 months of matched groups according to surgical intervention performed. Patients treated with clot evacuation had a significantly higher score on the modified Rankin Score (mRS; * *p <* 0.05) and lower score on the Glasgow Outcome Scale (GOS; ** p <* 0.05).

**Figure 3 biomedicines-12-01666-f003:**
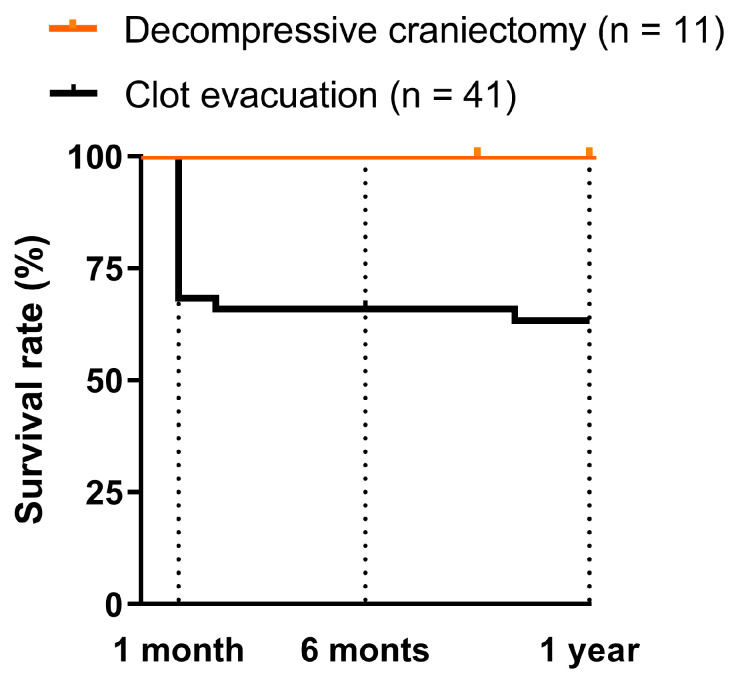
Mortality according to surgical intervention performed. One-month survival of patients with clot evacuation was 70%. This rate remained practically stable, falling to 67.5% at 6 months and to 64.9% at 1 year, while the survival of patients with decompressive craniectomy was 100%.

**Table 1 biomedicines-12-01666-t001:** Descriptive statistics and comparison of treatment groups.

	DC w/o CE	CE	Total	*p*-Value
Number of patients	11	18	29	
Intervention				
Clot evacuation	0 (0%)	13 (100%)	13 (59%)	<0.001
Decompressive craniectomy	9 (100%)	6 (46%)	15 (68%)	<0.05
Baseline characteristics				
Age (years)	49 ± 16 ^(1)^	57 ± 14	54 ± 15	0.238
Sex, male	60%	62%	59%	0.779
Hematoma characteristics				
Spontaneous etiology vs. trauma	78%	85%	82%	1.000
Aetiology by group				0.452
Spontaneous	11%	31%	23%	
HT ^(2)^	33%	15%	23%	
SAH ^(3)^	11%	31%	23%	
Amyloidosis	11%	8%	9%	
Anticoagulation	11%	0%	5%	
Reperfusion ^(4)^	22%	15%	18%	
Head injury	78%	46%	59%	0.297
Left laterality	89%	77%	82%	0.878
Subarachnoid component	11%	15%	14%	1.000
Open to ventricles	67%	54%	59%	0.873
Superficial location ^(5)^	33%	54%	46%	0.607
Sylvian location				
Capsule involvement	78%	54%	64%	0.486
External/extreme	33%	8%	18%	0.332
Putamen involvement	11%	15%	14%	1.000
Internal capsule involvement	11%	0%	5%	0.850
Caudate nucleus involvement	42 ± 15 ^(1)^	46 ± 18	44 ± 16	0.634
Volume ([A × B × C]/2, mL)	7.2 ± 1.6 ^(1)^	7.4 ± 1.7	7.3 ± 1.6	0.823
Midline shift (mm)				
Clinical characteristics				
GCS on admission	8 ± 1 ^(1)^	8 ± 2	8 ± 2	0.920
Surgery in <24 h ^(6)^	100%	85%	91%	0.631
Outcome				
Follow-up time (m)	8 (4–65) ^(7)^	24 (1–120)	18 (1–120)	0.337
Mortality	0%	23%	14%	0.358
GOS 4 + GOS 5	78%	15%	41%	<0.05
mRS 0, 1 or 2	78%	15%	41%	<0.05

^(1)^ Mean ± standard deviation, ^(2)^ hypertension, ^(3)^ spontaneous subarachnoid hemorrhage, ^(4)^ hemorrhage after ischemic stroke, ^(5)^ location < 1.5 cm from the cortex, ^(6)^ time from clinical deterioration to intervention, ^(7)^ median (range). DC w/o CE: decompressive craniectomy without clot evacuation; CE: clot evacuation.

**Table 2 biomedicines-12-01666-t002:** Logistic regression: factors associated with poor functional prognosis (GOS 1–3 or mRS 3–6).

Factor	OR (95% CI) ^a^	*p*-Value	Adjusted OR (95% CI) ^b^	*p*-Value
Decompressive craniectomy without clot evacuation	0.052 (0.01–0.46)	0.008	0.050 (0.04–0.61)	0.019
Age > 55 years	7.88 (1.11–56.12)	0.039	8.4 (0.7–105)	0.100
Sex, male	0.58 (0.10–3.40)	0.549		
Spontaneous etiology	0.80 (0.49–1.31)	0.378		
Left laterality	0.25 (0.04–1.66)	0.149		
Superficial location	0.58 (0.10–3.40)	0.549		
Volume (mL)	1.06 (0.99–1.14)	0.109		
Midline shift (mm)	1.25 (0.72–2.18)	0.436		
Glasgow Coma Scale on admission	0.74 (0.43–1.26)	0.264		

^a^ Crude OR obtained by univariate logistic regression; ^b^ adjusted OR by multivariate logistic regression; likelihood-ratio test 17.494; *p* < 0.001; degrees of freedom (DF) = 2. Goodness of fit: Homer–Lemeshow statistic = 1.17; DF = 2; *p* = 0.556 (no evidence of lack of fit). Area under the ROC curve = 0.107 (95% CI = 0.000 to 0.242).

## Data Availability

The raw data supporting the conclusions of this article will be made available by the authors upon request.

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
