# Peer review of "Decompressive Hemicraniectomy without Evacuation of Acute Intraparenchymal Hemorrhage"

_biomedicines, 2024, doi:10.3390/biomedicines12081666_

Round 1

Reviewer 1 Report

Comments and Suggestions for Authors

The manuscript titled "Decompressive hemicraniectomy without evacuation of acute intraparenchymal hemorrhage" presents an insightful study on the outcomes of decompressive craniectomy (DC) without clot evacuation in patients with acute intracerebral hemorrhage (ICH). This retrospective cohort analysis of 63 patients offers compelling evidence that DC without clot evacuation may yield better functional outcomes compared to the traditional approach of clot evacuation. The study is methodologically sound, utilizing matched univariate analyses and binary logistic regression to assess the results. The authors have clearly articulated the study’s rationale, methods, and findings, and the manuscript is well-organized. Notably, the study suggests a significant benefit in younger patients treated with DC without clot evacuation, which could influence future neurosurgical practices.

However, there are areas that require further attention. The retrospective design inherently carries a risk of selection bias, and the relatively small sample size, particularly after matching, might limit the broader applicability of the findings. The lengthy study period from 2007 to 2020 raises concerns about changes in medical practices that could have affected outcomes. The manuscript would benefit from a more in-depth discussion of these limitations and potential confounders. Additionally, while the statistical methods are robust, more transparency regarding the matching criteria and process would strengthen the study. Expanding the discussion to compare these results with existing literature would also provide a richer context and underscore the study’s contributions.

Given these points, I recommend the manuscript for minor revisions prior to acceptance. The authors should address the potential biases and elaborate on the study's limitations. Providing more details on the matching process and expanding the discussion to include a comparison with existing studies will enhance the manuscript’s impact. Once these revisions are made, the manuscript will be a valuable addition to neurosurgical literature, potentially guiding a shift in the management of acute ICH to improve patient outcomes.

Author Response

The manuscript titled "Decompressive hemicraniectomy without evacuation of acute intraparenchymal hemorrhage" presents an insightful study on the outcomes of decompressive craniectomy (DC) without clot evacuation in patients with acute intracerebral hemorrhage (ICH). This retrospective cohort analysis of 63 patients offers compelling evidence that DC without clot evacuation may yield better functional outcomes compared to the traditional approach of clot evacuation. The study is methodologically sound, utilizing matched univariate analyses and binary logistic regression to assess the results. The authors have clearly articulated the study’s rationale, methods, and findings, and the manuscript is well-organized. Notably, the study suggests a significant benefit in younger patients treated with DC without clot evacuation, which could influence future neurosurgical practices.

However, there are areas that require further attention. The retrospective design inherently carries a risk of selection bias, and the relatively small sample size, particularly after matching, might limit the broader applicability of the findings. The lengthy study period from 2007 to 2020 raises concerns about changes in medical practices that could have affected outcomes. The manuscript would benefit from a more in-depth discussion of these limitations and potential confounders. Additionally, while the statistical methods are robust, more transparency regarding the matching criteria and process would strengthen the study. Expanding the discussion to compare these results with existing literature would also provide a richer context and underscore the study’s contributions.

Given these points, I recommend the manuscript for minor revisions prior to acceptance. The authors should address the potential biases and elaborate on the study's limitations. Providing more details on the matching process and expanding the discussion to include a comparison with existing studies will enhance the manuscript’s impact. Once these revisions are made, the manuscript will be a valuable addition to neurosurgical literature, potentially guiding a shift in the management of acute ICH to improve patient outcomes.

  • Many thanks to the reviewer for his good considerations, which we find interesting and will contribute to the improvement of the manuscript. Certainly, this is a follow-up of many years, which may entail inherent selection bias. However, this is a fact that has been taken into account.
  • The surgical technique has not been modified and both groups are homogenized during this period. The time from the “Stroke” to the surgical treatment has been homogeneous. On the other hand, both techniques have persisted to this day, precisely because the literature is still unclear as to which approach is more appropriate. In our study, none of the techniques prevailed over the other in the years in which the series was collected.
  • We understand that the number of patients is small. The difficulty of finding “the different types of hemorrhages” in each group and comparing them in age, location, etiology, (without the presence of an extra-axial component) that could alter the results, is not easy.
  • In the study carried out, we intend to find the respective matching for each type of hemorrhage, a complete homogenization between the respective groups and subgroups.
  • Following the reviewer's suggestions, we have added a section on weaknesses and strengths and have also added more information in the discussion. This new information is highlighted in yellow.

Reviewer 2 Report

Comments and Suggestions for Authors

Firstly, I appreciate your article for its aim to clarify a topic widely debated in the literature. This research represents a valuable contribution to the field. However, I have some questions.

1.      Can you describe the inclusion and exclusion criteria in more detail, particularly the reasons for excluding certain patients. It seems that the hematoma was evacuated in cases where it caused a shift. Is this correct? In which situations would you recommend hematoma evacuation?

2.      Could you please discuss the results of any subgroup analyses, such as those based on age or hematoma volume?

3.      Could you kindly elaborate on the study's limitations, including its retrospective nature, small sample size, and potential selection biases?

Author Response

Firstly, I appreciate your article for its aim to clarify a topic widely debated in the literature. This research represents a valuable contribution to the field. However, I have some questions.

  1. Can you describe the inclusion and exclusion criteria in more detail, particularly the reasons for excluding certain patients. It seems that the hematoma was evacuated in cases where it caused a shift. Is this correct? In which situations would you recommend hematoma evacuation?
  • Further information on inclusion and exclusion criteria has been added in section 2. Material and Methods; Patient Selection. 

    This new information is highlighted in yellow.

  • In general, the inclusion criteria are based on a previous good baseline status on the Karnosky EK scale >70, with hematomas >30ml. No evidence of pupillary alteration prior to treatment. Time of surgical indication since the stroke (minor < 24 hours). Signs of elevated intracranial pressure, such as headache, vomiting and/or altered consciousness.
  • Exclusion criteria. EK < 70 prior to surgical treatment. Presence of comorbidities (such as Cancer, effects on other organs or systems including trauma or other diseases that could modify the patient's condition), as well as severe pre-existing neurological deficits, as the potential for significant recovery may be limited.
  • In addition to the above, it must be taken into account that we were able to control the variables of both groups and they were homogenized, both in age, sex, laterality of the hematoma, volume, intraparenchymal involvement, opening to the ventricular system, and risk factors that could alter the evolution, including time until surgical treatment. Perhaps this is one of the great strengths of our work.

  1. Could you please discuss the results of any subgroup analyses, such as those based on age or hematoma volume?
  • The age of the groups was established as under 70 years of age, although initially we followed the recommendations of the literature that, up to the age of 80, certain benefits of neurosurgical treatment can be found. The impossibility of finding similar patients, at ages closer to 80 years, as well as younger patients, only allowed us to take into account the age of the study patients.
  • Regarding the volume of the hematoma, our study is based mainly on volumes of 30-60ml. No patient with hematomas smaller than 30ml or larger than 60ml was included. In relation to those smaller than 30ml, we think that medical treatment may be one of the best initial and most plausible alternatives. Hematomas of larger volumes have an unfavorable functional prognosis. In our opinion, the groups that benefit the most from surgical treatment are patients with hematoma volumes that are between 30-60ml. For this reason, we try to clarify which of both surgical techniques can offer the best clinical and functional results.
  • In this study, we did not present patients who did not have a counterpart with similar clinical radiological characteristics, since this could alter the results to the benefit of one group or another.

  1. Could you kindly elaborate on the study's limitations, including its retrospective nature, small sample size, and potential selection biases?
  • As we have responded to the previous referee, we understand that the sample is small. The difficulty of finding “the different types of hemorrhages” in each group and comparing them in age, location, etiology, (without the presence of an extra-axial component) that could alter the results, is not easy. Furthermore, this involves a follow-up of many years, which may lead to inherent selection bias. However, this is a fact that has been taken into account.
  • We have added a section on limitations and strengths of the study in the manuscript. 

    This new information is highlighted in yellow.

Reviewer 3 Report

Comments and Suggestions for Authors

Dear Authors, 

Thank you for conducting this paper. 

Although the title and the content of the study are interesting, the paper lacks presentation and a limited number of cases as well. 

Please rewrite the introduction. Remove the additional details that have been mentioned in the method section. Please add limitations of the study and highlight the weak points of the study.  

Sincerely 

Comments on the Quality of English Language

There are some grammatical errors throughout the paper that should be modified.

Author Response

Dear Authors, 

Thank you for conducting this paper. 

Although the title and the content of the study are interesting, the paper lacks presentation and a limited number of cases as well. 

Please rewrite the introduction. Remove the additional details that have been mentioned in the method section. Please add limitations of the study and highlight the weak points of the study.  

Sincerely 

There are some grammatical errors throughout the paper that should be modified.

  • Thank you very much, we accept your considerations, which are interesting and relevant.
  • The number of cases is limited, this being one of our weaknesses. Certainly, in our center we have an average of 75 (Stroke) with intraparenchymal cerebral hemorrhages per year. However, trying to homogenize hematoma groups is not easy. From the type of impact, comorbidities, etiology and time to surgical indication, they are crucial to obtain reproducible results. Without taking into account the clinical-radiological characteristics of each patient with hematoma. With the study we try to clarify the most appropriate surgical treatment for this type of patient, which compromises their vital and functional prognosis.
  • Following the referee's suggestions, we have modified the instruction and added a limitations section. We have also checked for grammatical errors.

Round 2

Reviewer 3 Report

Comments and Suggestions for Authors

Dear Authors, 

Thank you again for conducting this paper. 

I have carefully read the paper. The revised format of the paper has been enhanced in both presentation and scientific aspects. 

Thank you 

Sincerely 

Author Response

Thanks to the reviewer for his comments